# Testosterone Therapy and Diaphragm Performance in a Male Patient with COVID-19: A Case Report

**DOI:** 10.3390/diagnostics12020535

**Published:** 2022-02-19

**Authors:** Gloria Martins, Juan Carlos Rosso Verdeal, Helio Tostes, Alice Ramos Oliveira da Silva, Bernardo Tessarollo, Nazareth Novaes Rocha, Patricia Rieken Macedo Rocco, Pedro Leme Silva

**Affiliations:** 1Laboratory of Pulmonary Investigation, Institute of Biophysics Carlos Chagas Filho, Federal University of Rio de Janeiro, Rio de Janeiro 21941-599, Brazil; gloria.martins@gmail.com (G.M.); nn_rocha@hotmail.com (N.N.R.); prmrocco@gmail.com (P.R.M.R.); 2Barra D’Or, Rio de Janeiro 22775-002, Brazil; jverdeal@gmail.com (J.C.R.V.); htostesf@gmail.com (H.T.); aliceramosoliveiradasilva@gmail.com (A.R.O.d.S.); btessa@gmail.com (B.T.); 3Department of Physiology, Fluminense Federal University, Niterói 24210-130, Brazil

**Keywords:** testosterone, diaphragm, COVID-19

## Abstract

Low levels of testosterone may lead to reduced diaphragm excursion and inspiratory time during COVID-19 infection. We report the case of a 38-year-old man with a positive result on a reverse transcriptase-polymerase chain reaction test for SARS-CoV-2, admitted to the intensive care unit with acute respiratory failure. After several days on mechanical ventilation and use of rescue therapies, during the weaning phase, the patient presented dyspnea associated with low diaphragm performance (diaphragm thickness fraction, amplitude, and the excursion-time index during inspiration were 37%, 1.7 cm, and 2.6 cm/s, respectively) by ultrasonography and reduced testosterone levels (total testosterone, bioavailable testosterone and sex hormone binding globulin (SHBG) levels were 9.3 ng/dL, 5.8 ng/dL, and 10.5 nmol/L, respectively). Testosterone was administered three times 2 weeks apart (testosterone undecanoate 1000 mg/4 mL intramuscularly). Diaphragm performance improved significantly (diaphragm thickness fraction, amplitude, and the excursion-time index during inspiration were 70%, 2.4 cm, and 3.0 cm/s, respectively) 45 and 75 days after the first dose of testosterone. No adverse events were observed, although monitoring was required after testosterone administration. Testosterone replacement therapy led to good diaphragm performance in a male patient with COVID-19. This should be interpreted with caution due to the exploratory nature of the study.

## 1. Introduction

Cross-sectional [1] and case–control [2] studies suggest an association between low testosterone levels and a systemic increase in inflammatory cells and mediators (neutrophils, procalcitonin, and ferritin) in cases of pneumonia induced by SARS-CoV-2. Reduced testosterone levels may be the result of a complex interaction between the host and the virus. During the hyperinflammatory acute phase, deregulation of gonadal function may occur, leading in some cases to transient hypogonadism [3,4,5]. In addition, low levels of testosterone may have a negative impact on skeletal muscle performance, particularly on the diaphragm. Diaphragm performance can be defined according to the excursion-time (E-T) index [6]. The E-T index takes into account diaphragm excursion, measured by ultrasonography, as a surrogate for mean tidal pressure (PI) generated by respiratory muscle multiplied by inspiratory time (TI) and as a proxy for the work performed by the diaphragm. Neuromuscular disorders may evolve in patients with COVID-19 [7], which may be caused by both the release of proinflammatory mediators [8] and the frequent use of myopathogenic medications, such as corticosteroid and neuromuscular blockers during the acute phase [9], as well as immobility. Testosterone supplementation can increase lean body mass [10], especially when combined with growth hormone [11]. In COVID-19, some studies have shown that the proinflammatory state that arises due to low testosterone levels can be suppressed with the provision of exogenous testosterone [12,13]. Thus, it remains unclear whether testosterone administration can have a clinical impact on skeletal muscle performance, mainly the diaphragm, during the weaning process after the acute phase of COVID-19.

## 2. Case Study

In carrying out this study, we followed the CARE guidelines (https://www.care-statement.org/ (accessed on 10 January 2022)). Written informed consent was obtained from the patient to publish this paper. On 15 January 2021, a 38-year-old man with body mass index of 35 kg/m^2^, with a positive result on a reverse transcriptase-polymerase chain reaction (RT-PCR) test for SARS-CoV-2 infection, was admitted to the intensive care unit (ICU) with acute respiratory failure. The patient was promptly intubated endotracheally and mechanically ventilated. After infusion with rocuronium bromide (8 μg/kg/min), his PaO_2_/FiO_2_ (the ratio of arterial oxygen partial pressure to fractional inspired oxygen) was 83 mmHg, and a chest computed tomography scan showed bilateral diffuse consolidations and ground glass opacities over 50% of the total pulmonary area. No personal or family medical history of rhabdomyolysis, myoglobinuria, or any type of muscle disease was reported. The patient did not take statins or any myotoxic agents. The patient rapidly progressed to multiple organ dysfunction with renal replacement therapy, prone position ventilation, and nitric oxide inhalation. No clinical and oxygenation improvements were observed; therefore, extracorporeal membrane oxygenation (ECMO) was initiated. The patient was kept in ECMO for 12 days, followed by 19 days with a neuromuscular blocking agent (rocuronium bromide). In addition, the patient received midazolam and fentanyl for 43 days, dexmedetomidine for 6 days, methylprednisolone for 10 days, and amikacin sulfate for 30 days.

After sedation, anesthesia, and the neuromuscular blocking agent were withdrawn, the patient presented dyspnea associated with myalgia and fatigue, as well as severe proximal muscle weakness in both upper and lower limbs on clinical examination. Evaluation of segmental muscle strength showed diffuse and symmetrical weakness, as well as deep and superficial sensory disturbance. Flaccid tetraparesis and limb muscle atrophy were also observed. Creatine kinase level was not increased on laboratory examination. Neuroradiologic examinations were normal. Ultrasonography was used to measure (1) diaphragm muscle thickness fraction (thickness at inspiration − thickness at expiration/thickness at expiration) × 100; (2) diaphragm amplitude excursion; (3) excursion-time index [6] during inspiration; and (4) rectus femoris muscle size and quality, as an internal control parameter. Using a validated technique, right diaphragm muscle thickness was measured using a high-frequency (13 MHz) linear array transducer in the zone of apposition (assessed at 0.5–2 cm below the costophrenic sinus) between the anterior and midaxillary lines at the level of the 9th or 10th intercostal space [14]. The location of the placement of the ultrasonographic probe was marked to enhance consistency of the day-to-day measurements. The thickness of expiratory and inspiratory diaphragm muscle was measured on 2 consecutive breaths from 2 separate images. Measurements were repeated by the same investigator to ensure comparable measurements (defined as <10% difference); the mean of all 4 measurements was used for analysis. On 23 February 2021, defined as the baseline, the patient was breathing spontaneously with supplemental oxygen by tracheostomy; diaphragm thickness fraction, amplitude, and the excursion-time index during inspiration were 37%, 1.7 cm, and 2.6 cm/s, respectively. At this time, total testosterone, bioavailable testosterone and sex hormone binding globulin (SHBG) levels were 9.3 ng/dL, 5.8 ng/dL, and 10.5 nmol/L, respectively (Table 1). In addition, C-reactive protein and ferritin levels were 6.6 mg/dL and 1996 ng/mL, respectively. Testosterone was administered 3 times 2 weeks apart (testosterone undecanoate 1000 mg/4 mL intramuscularly; NEBIDO, Bayer, Germany). Forty-five days after the first dose, the patient was decannulated and breathed spontaneously in room air. Table 1 and Figure 1 show the diaphragm thickness fraction, amplitude, and excursion-time index during inspiration, which improved after 45 days and sustained up to 75 days after the first dose of testosterone. Accordingly, total testosterone, bioavailable testosterone, and SHBG levels increased after 45 days and remained increased 75 days after the first dose of testosterone, just before hospital discharge. A progressive decrease in C-reactive protein and ferritin levels from baseline to 75 days after the first dose of testosterone can be observed. No major differences were observed in the size and quality of rectus femoris muscle over time.

Thickness fraction = (thickness at inspiration − thickness at expiration)/thickness at expiration × 100. Baseline was defined as the time when the patient was breathing spontaneously with supplemental oxygen by tracheostomy. After 45 and 75 days, after the first dose of testosterone administration, testosterone was administered 3 times 2 weeks apart (testosterone undecanoate 1000 mg/4 mL intramuscularly; NEBIDO, Bayer, Germany). Normal total testosterone range (males ≥ 19 years), 240–950 ng/dL; normal bioavailable testosterone range (males 30–39 years), 72–235 ng/dL; normal sex hormone binding globulin range (males), 10–57 nmol/L; normal total prostate-specific antigen, below 4 ng/mL; normal free prostate-specific antigen range (males 40–49 years), 0–2.5 ng/mL; normal C-reactive protein range, 0.8–100 mg/dL; normal ferritin range, 20–250 ng/mL.

## 3. Discussion

COVID-19 significantly affects more men than women [15]; however, there are controversies regarding the mechanisms. In this line, men are more prone to co-morbidities (e.g., smoke exposure) compared to women, even before the SARS-CoV2 pandemic [16]. In contrast, the X chromosome contains several genes related to viral infections response, such as the toll-like receptor 7, which may confer innate immune protection in women [17]. Transmembrane protease serine 2 (TMPRSS2) is an androgen receptor, which together with angiotensin-converting enzyme (ACE)-2 promotes SARS-CoV2 invasion of host cells [18]. TMPRSS2 is also the most frequently altered gene in primary prostate cancer [19] and its expression is positively regulated by androgens, which may increase TMPRSS2 expression in the surface of bronchial respiratory cells and promote a wide gate to host infection by SARS-CoV2 [20]. Theoretically, men with low levels of androgens would be protected against SARS-CoV2 infection, but this is not observed. Cross-sectional [1] and case–control [2] studies suggest an association between low testosterone levels and blood inflammatory cells and markers of the severity of SARS-CoV-2 pneumonia, such as neutrophils, procalcitonin, and ferritin. The inflammatory cytokine storm may induce muscle weakness by at least four processes: (1) initial myofibrillar protein cleavage by activated caspase-3; (2) degradation by the ubiquitin–proteosome system; (3) interruption of muscle protein synthesis through inhibition of mammalian target of rapamycin complex 1 (mTORc1), which impairs muscle turnover; and (4) deregulation of muscle growth hormones, such as insulin-like growth factor (IGF)-1.

Testosterone supplementation can increase lean body mass, although a recent clinical trial showed no effect on lower-body muscle function [10]. Nevertheless, testosterone combined with growth hormone resulted in substantial gains in lean mass, voluntary muscle strength, and aerobic endurance compared with testosterone alone [11]. The increase in muscle function may depend on the deficits that occurred before intervention, in this case, exogenous testosterone administration. In COVID-19, some studies have shown that the proinflammatory state that arises due to low testosterone levels can be suppressed with the provision of exogenous testosterone [12,13]. Here, we observed high levels of C-reactive protein and ferritin associated with reduced levels of total and bioavailable testosterone at the baseline. In addition, the low total and bioavailable testosterone and SHBG levels observed at baseline may be influenced by high BMI [21]. Although it is difficult to infer causality, after administration of exogenous testosterone, C-reactive protein and ferritin levels were reduced and total and bioavailable testosterone levels increased, which may have contributed to improved muscle function. For instance, keeping testosterone levels within the physiologic range has been shown to have a protective effect on forced expiratory volume and forced vital capacity [22]. In elderly patients with chronic heart failure, who are a frail population [23], Caminiti et al. found that testosterone replacement therapy improved peak oxygen consumption and respiratory function. We may hypothesize that an equivalent effect would occur in patients with COVID-19 who experience great skeletal muscle function deficits after ICU admission.

Exogenous administration of testosterone needs to be closely monitored because it may exacerbate benign prostatic hyperplasia and prostate cancer. No changes in prostate-specific antigen levels were observed (Table 1), and albumin levels were within 3.0 to 3.3 g/dL during the time course. Furthermore, in patients with COVID-19, exogenous administration of testosterone can increase the risk of thrombosis [24], and therefore should not be administered to patients with known thromboembolic events. Here, leg and arm Doppler evaluation did not reveal thrombosis during the time course.

Exogenous administration of testosterone led to better diaphragm function in a male patient with COVID-19. Due to the nature of the study, causality was very low. However, this case report is a step forward for designing further clinical studies

## 4. Conclusions

Testosterone replacement therapy led to good diaphragm performance in a male patient with COVID-19. This should be interpreted with caution due to the exploratory nature of the study.

## Figures and Tables

**Figure 1 diagnostics-12-00535-f001:**
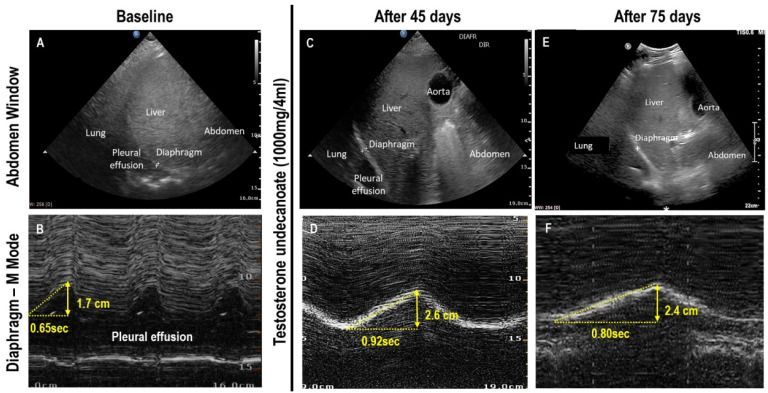
(**A**) Abdomen window at baseline showing extensive pleural effusion. (**B**) Diaphragm M-mode at baseline; diaphragm amplitude during inspiration = 1.7 cm; diaphragm thickness fraction = 37%; excursion-time index = 2.6 cm/s. Testosterone was administered 3 times 2 weeks apart (testosterone undecanoate 1000 mg/4 mL intramuscularly; NEBIDO, Bayer, Germany). (**C**) Abdomen window after 45 days. (**D**) Diaphragm M-mode after 45 days; diaphragm amplitude during inspiration = 2.6 cm; diaphragm thickness fraction = 60%; excursion-time index = 2.4 cm/s. (**E**) Abdomen window after 75 days. (**F**) Diaphragm M-mode after 75 days; diaphragm amplitude during inspiration = 2.4 cm; diaphragm thickness fraction = 50%; excursion-time index = 3.0 cm/s.

**Table 1 diagnostics-12-00535-t001:** Diaphragm ultrasonography and blood laboratory measurements over time.

	Baseline	After 45 Days	After 75 Days
**Diaphragm ultrasonography measurements**			
Thickness at expiration (cm)	0.30	0.31	0.30
Thickness at inspiration (cm)	0.41	0.51	0.51
Thickness fraction (%)	37	67	70
Excursion-time index during inspiration (cm/s)	2.6	2.4	3.0
**Blood laboratory measurements**			
Total testosterone (ng/dL)	9.3	212	332
Bioavailable testosterone (ng/dL)	5.8	129	196
Sex hormone binding globulin (nmol/L)	10.5	15.5	17.9
Total prostate-specific antigen (ng/mL)	0.5	0.5	0.4
Free prostate-specific antigen (ng/mL)	0.2	0.2	0.2
C-reactive protein (mg/dL)	6.6	5.4	2.2
Ferritin (ng/mL)	1996	1983	1041

## Data Availability

The datasets analyzed for this study are available under communication with Author and Corresponding Author.

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
