# Peer review of "Testosterone Therapy and Diaphragm Performance in a Male Patient with COVID-19: A Case Report"

_diagnostics, 2022, doi:10.3390/diagnostics12020535_

Round 1
Reviewer 1 Report
- General comment (originality, scientific accuracy, strengths and/or weaknesses):
The manuscript entitled “Testosterone Therapy and Diaphragm Performance in a Male Patient with COVID-19: A Case Report” report the case of an obese male affected by COVID-19 and treated with testosterone in order to improve diaphragm performance. The case report is well written, understandable and provide an interesting hypothesis that should be further evaluated in proper studies. Few corrections and clarifications should be reported, in order to improve the quality of manuscript presentation.
- Major Corrections
Introduction
28: Regarding this issue see also: doi: 10.22074/ijfs.2020.6302 and https://doi.org/10.1016/j.mehy.2020.110287
44: the elevated body mass index could heavily influence outcomes and testosterone levels and should be reported in the discussion
Discussion
132: However, this could be related to dose and formulations as reported in https://doi.org/10.1210/jc.2008-2338
Conlusions:
Considering the possible hypotheses, conclusions seems a bit too rushed.
- Minor Corrections
Abstract
20: add keywords
Introduction
57: the prolonged immobility and the 19 days of neuromuscular blocking agent could also have influenced the decreased strength of diaphragm
Discussion
115: briefly report those differences
151: add citation
153: close the discussion with some considerations on possible hypotheses and future possibilities of research
Author Response
Response to Reviewer #1
Comments and Suggestions for Authors
General comment (originality, scientific accuracy, strengths and/or weaknesses):
The manuscript entitled “Testosterone Therapy and Diaphragm Performance in a Male Patient with COVID-19: A Case Report” report the case of an obese male affected by COVID-19 and treated with testosterone in order to improve diaphragm performance. The case report is well written, understandable and provide an interesting hypothesis that should be further evaluated in proper studies. Few corrections and clarifications should be reported, in order to improve the quality of manuscript presentation.
Response: Thank you very much for the positive comments.
Major Corrections
Introduction
28: Regarding this issue see also: doi: 10.22074/ijfs.2020.6302 and https://doi.org/10.1016/j.mehy.2020.110287
Response: Thank you for the suggestions of these two papers. We have now included both references in the revised version of the manuscript.
44: the elevated body mass index could heavily influence outcomes and testosterone levels and should be reported in the discussion
Response: Thank you for the suggestions. Indeed, this is an important issue. We have now added a new sentence to better explain the low levels of testosterone presented by the patient at baseline: “In addition, the low total and bioavailable testosterone and SHBG levels observed at baseline may be influenced by high BMI (Diaz-Arjonilla M, IJIR 2009).”
Discussion
132: However, this could be related to dose and formulations as reported in https://doi.org/10.1210/jc.2008-2338
Response: Thank you for the comments. We have now included this study to highlight that testosterone combined with growth hormone may lead to additional benefits compared to testosterone alone: “Nevertheless, testosterone combined with growth hormone resulted in substantial gains in lean mass, voluntary muscle strength, and aerobic endurance compared with testosterone alone (Sattler FR, J Clin Endocrinol Metab 2009).”
Conclusions:
Considering the possible hypotheses, conclusions seems a bit too rushed.
Response: We have now rewritten the conclusion: “Testosterone replacement therapy led to good diaphragm performance in a male patient with COVID-19. This should be interpreted with caution due to exploratory nature of the study.”
Minor Corrections
Abstract
20: add keywords
Response: Key words have now been added to the revised version of the manuscript.
Introduction
57: the prolonged immobility and the 19 days of neuromuscular blocking agent could also have influenced the decreased strength of diaphragm
Response: Indeed, we agree with the Reviewer. In the introduction, we have highlighted the role of neuromuscular blocking in neuromuscular disorders (line 37, ref #7). Moreover, we have now included the immobility as an additional cause of neuromuscular disorders.
Discussion
115: briefly report those differences
Response: Thank you for this suggestion. We have now briefly exposed the mechanisms and factors that may explain why men are more affected than women: “In this line, men are more prone to co-morbidities (e.g. smoke exposure) compared to women even before the SARS-CoV2 pandemia. (Eshima N, PLoS One 2011). In contrast, X chromosome contains several genes related to viral infections response, such as toll-like receptor 7, which may confer innate immune protection in women (Bienvenu LA Cardiovascular Res 2020).”
151: add citation
Response: The citation (10.1001/jamainternmed.2019.5135) has now been added.
153: close the discussion with some considerations on possible hypotheses and future possibilities of research
Response: Thank you for the comments. We have now included the following sentence: “Exogenous administration of testosterone led to better diaphragm function in a male patient with COVID-19. Due to the nature of the study, causality is very low. However, this case report is a step forward for designing further clinical studies.”
Reviewer 2 Report
In this manuscript, several concerns need to be addressed as follows:
1. The keywords are missed.
2. The abstract needs to be provided with an informative conclusion.
3. The introduction is very concise and more details on the testosterone administration protocols should be clarified. Please clarify if the administered dose of testosterone is therapeutically approved in all treatment regimes.
4. Line 54: What the author means with nitric oxide inhalation?
5. Table 1 legend needs to be more informative with details of testosterone administration.
6. Lines 135-137: some studies but two only references exist. At least, three references to such studies should be added.
7. Lines 149-150: the authors mentioned that "No changes in prostate-specific antigen levels were observed, and albumin levels were within 3.0 to 3.3 g/dL". However, they have not given any information on these biochemical indicators during the presentation of the case.
8. Line 152: AND. Please, revise.
9. The conclusion is very concise and further perspectives should be added.
Author Response
Response to Reviewer #2
In this manuscript, several concerns need to be addressed as follows:
- The keywords are missed.
Response: Key words have now been added to the revised version of the manuscript.
The abstract needs to be provided with an informative conclusion.
Response: Thank you for this suggestion. An informative conclusion has now been added.
- The introduction is very concise and more details on the testosterone administration protocols should be clarified. Please clarify if the administered dose of testosterone is therapeutically approved in all treatment regimes.
Response: This is an important comment. We have now modified the introduction and included the following sentences regarding testosterone replacement in overall and COVID-19 population. “Testosterone supplementation can increase lean body mass although a recent clinical trial showed no effect on lower-body muscle function [13]. Nevertheless, testosterone combined with growth hormone resulted in substantial gains in lean mass, voluntary muscle strength, and aerobic endurance compared with testosterone alone (Sattler FR, J Clin Endocrinol Metab 2009). The increase in muscle function may depend on the deficits that occurred before intervention, in this case, exogenous testosterone administration. In COVID-19, some studies have shown that the proinflammatory state that arises due to low testosterone levels can be suppressed with the provision of exogenous testosterone [14,15].”
The administered dose of testosterone (testosterone undecanoate 1000 mg/4 mL intramuscularly; NEBIDO, Bayer, Germany) is approved for clinical use.
- Line 54: What the author means with nitric oxide inhalation?
Response: Inhaled nitric oxide has been used as a rescue therapy for COVID-19 acute respiratory distress syndrome to improve oxygenation.
- Table 1 legend needs to be more informative with details of testosterone administration.
Response: We have now added details of testosterone administration in the legend. “Testosterone was administered 3 times 2 weeks apart (testosterone undecanoate 1000 mg/4 mL intramuscularly; NEBIDO, Bayer, Germany).”
- Lines 135-137: some studies but two only references exist. At least, three references to such studies should be added.
Response: Thank you for these comments. We included one additional clinical trial, as it follows: Malkin CJ et al. J Clin Endocrinol Metab. 2004;89(7):3313–3318. Thus, three studies as total.
- Lines 149-150: the authors mentioned that "No changes in prostate-specific antigen levels were observed, and albumin levels were within 3.0 to 3.3 g/dL". However, they have not given any information on these biochemical indicators during the presentation of the case.
Response: The Authors thank the Reviewer about this comment. We added the total and free prostate-specific antigen levels in Table 1.
- Line 152: AND. Please, revise.
Response: Typo revised.
- The conclusion is very concise and further perspectives should be added.
Response: The Authors thank the Reviewer for this comment. This was also a comment from Reviewer #1. According to your comments and Reviewer #1, we have now added further perspectives and detailed conclusion.
Further perspectives
“Exogenous administration of testosterone led to better diaphragm function in a male patient with COVID-19. Due to the nature of the study, causality is very low. However, this case report is a step forward for designing further clinical studies.”
Detailed conclusion
“Testosterone replacement therapy led to good diaphragm performance in a male patient with COVID-19. This should be interpreted with caution due to exploratory nature of the study.”
Round 2
Reviewer 2 Report
The authors addressed all comments